# scBoolSeq: Linking scRNA-seq statistics and Boolean dynamics

**Gustavo Magaña-López**[1], **Laurence Calzone**[2,3,4], **Andrei Zinovyev**[5], **Loïc Paulevé**[1]*

**1** Univ. Bordeaux, CNRS, Bordeaux INP, LaBRI, UMR 5800, Talence, France, **2** Institut Curie, Université PSL, Paris, France, **3** INSERM, U900, Paris, France, **4** Mines ParisTech, Université PSL, Paris, France, **5** In silico R&D, Evotec, Toulouse, France

* loic.pauleve@labri.fr

**Data Availability Statement:** Source code is available at https://github.com/bnediction/scBoolSeq. Notebooks for demonstrating scBoolSeq usage and reproducing the case studies presented in this paper can be visualised and

## Abstract

Boolean networks are largely employed to model the qualitative dynamics of cell fate processes by describing the change of binary activation states of genes and transcription factors with time. Being able to bridge such qualitative states with quantitative measurements of gene expression in cells, as scRNA-seq, is a cornerstone for data-driven model construction and validation. On one hand, scRNA-seq binarisation is a key step for inferring and validating Boolean models. On the other hand, the generation of synthetic scRNA-seq data from baseline Boolean models provides an important asset to benchmark inference methods. However, linking characteristics of scRNA-seq datasets, including dropout events, with Boolean states is a challenging task.

We present scBoolSeq, a method for the bidirectional linking of scRNA-seq data and Boolean activation state of genes. Given a reference scRNA-seq dataset, scBoolSeq computes statistical criteria to classify the empirical gene pseudocount distributions as either unimodal, bimodal, or zero-inflated, and fit a probabilistic model of dropouts, with gene-dependent parameters. From these learnt distributions, scBoolSeq can perform both binarisation of scRNA-seq datasets, and generate synthetic scRNA-seq datasets from Boolean traces, as issued from Boolean networks, using biased sampling and dropout simulation. We present a case study demonstrating the application of scBoolSeq's binarisation scheme in data-driven model inference. Furthermore, we compare synthetic scRNA-seq data generated by scBoolSeq with BoolODE's, data for the same Boolean Network model. The comparison shows that our method better reproduces the statistics of real scRNA-seq datasets, such as the mean-variance and mean-dropout relationships while exhibiting clearly defined trajectories in two-dimensional projections of the data.

## Author summary

The qualitative and logical modelling of cell dynamics has brought precious insight into gene regulatory mechanisms that drive cellular differentiation and fate decisions by predicting cellular trajectories and mutations for their control. However, the design and validation of these models is impeded by the quantitative nature of experimental

downloaded at https://nbviewer.org/github/bnediction/scBoolSeq-supplementary. They are archived at https://doi.org/10.5281/zenodo.12582904.

**Funding:** Work of GML and LP was partly supported by the French Agence Nationale pour la Recherche (ANR) in the scope of the project "BNeDiction" (grant number ANR-20-CE45-0001). Work of GML was partly supported by the Talentos de Exportación - JuventudEsGto scholarship program of the Mexican State of Guanajuato. Work of LP was partly supported by the French government in the scope of France 2030 project "AI4scMED" operated by ANR (grant number ANR-22-PESN-0002). LC was partly supported by ModICeD project from MIC ITMO 2020. AZ, LC were supported by ANR as part of the "Investissements d'avenir" program, reference ANR-19-P3IA-0001 (PRAIRIE 3IA Institute). The funders had no role in study design, data collection and analysis, decision to publish, or preparation of the manuscript.

**Competing interests:** The authors have declared that no competing interests exist.

measurements of cellular states. In this paper, we provide and assess a new methodology, scBoolSeq for bridging single-cell level pseudocounts of RNA transcripts with Boolean classification of gene activity levels. Our method, implemented as a Python package, enables both to *binarise* scRNA-seq data in order to match quantitative measurements with states of logical models, and to generate synthetic data from Boolean traces to benchmark inference methods. We show that scBoolSeq accurately captures the main statistical features of scRNA-seq data, including measurement dropouts, improving significantly the state of the art. Overall, scBoolSeq brings a statistically-grounded method for enabling the inference and validation of qualitative models from scRNA-seq data.

## Introduction

Unveiling the mechanisms that regulate cellular decisions is a central task in systems biology. For instance, numerous efforts have been conducted to elucidate the core mechanisms that control differentiation and cell fate decision processes such as osteogenesis [1–3], haematopoiesis [4–7], dopaminergic neuron differentiation [8], early retinal development [9], and various cancer types [10–13].

The advent of single-cell RNA sequencing (scRNA-seq) technologies has greatly enhanced the resolution with which these dynamic phenomena can be studied. As a preliminary step, most studies first determine cell identities via either clustering and subsequent manual annotation or via the direct classification of cells [14]. Furthermore, trajectory reconstruction methods [15–17] allow visualising and hypothesising how gradual changes in gene expression eventually lead to a commitment to specific lineages and phenotypes. A tremendous challenge is then to identify regulatory mechanisms that control the identified dynamics of expression patterns and ultimately phenotypes.

Boolean networks are widely employed to model cellular differentiation [18–21] and fate decision [22, 23]. In these models, the activity of biological entities is represented as either active or inactive. This coarse-grained view of gene expression levels helps counter the varying levels of technical noise caused by sequencing technologies. The binary representation allows reasoning on the causal relationships between entities without having to estimate kinetic parameters or regulation thresholds while ensuring consistency with underlying quantitative models [24]. Boolean models can predict trajectories and conclude on the impossibility of certain behaviours, optionally subject to mutations, and can encompass thousands of genes. They revealed to be a powerful and relevant modelling approach to predict combinations of genetic perturbations to control cell fate decision [25, 26].

Nevertheless, linking qualitative gene activation states with their quantitative measurements, such as count of RNA transcripts, is a delicate task with high stakes for Boolean modelling. We present scBoolSeq which, given a reference dataset, provides a bidirectional link between scRNA-seq and Boolean activation states.

The binary coarse-graining of scRNA-seq, we refer to as *binarisation*, consists in assigning a qualitative active or inactive state to a gene, from one single-cell or a pool of single-cell measurements. The pools of cells usually correspond to phenotypes and other important cellular states. As Boolean models aim at predicting stability and traces between such cellular states, binarised data are crucial to assess their fitness with traces and steady states. One can easily note that the binary classification may be irrelevant in some cases, e.g., when in intermediate activation levels, or because of lacking statistical support. Therefore, it is important that binarisation methods result in three possible outcomes for the gene's state: active, inactive, or

undetermined. However, numerous methods fully binarise transcriptome data with no regard for uncertainty or intermediate expression and the diversity of empirical pseudocount distributions [27]. REFBOOL [28] provided an important effort for quantifying statistical uncertainty for the binarisation and allowing intermediate states. Their approach aims at exploiting a user-defined gene expression library which serves as a proxy to take into consideration the context of the global gene expression landscape when coarse-graining data. Unfortunately, this approach is only available for bulk RNA-seq data.

The inverse operation of binarisation consists in generating RNA pseudocounts from Boolean activation states. Coupled with simulations of Boolean models, this enables the generation of synthetic datasets from Boolean models subject to ranges of combinations of perturbations, simulating gene knock-out or constitutive activation, for instance. The resulting synthetic scRNA-seq data can then serve as a basis to evaluate inference methods, such as gene regulatory networks inference, trajectory inference, and Boolean model inference.

Generating single-cell and bulk RNA-Seq data has been addressed by count simulators [29–31]. With different underlying assumptions, count simulators reproduce the statistical characteristics of real datasets via parametric and semi-parametric approaches. They are capable of simulating a wide variety of scenarios and even batch effects, but generally fail at integrating information from Gene Regulatory Network (GRN) known a priori. Efforts have been made to integrate knowledge about GRNs into count simulators [32]. However, this method requires the GRN to be a directed acyclic graph, which might not be the case in general. Alternative methods rely on translating Boolean networks into non-linear Ordinary Differential Equations (ODEs). A first work in this line was ODEFY which presented a canonical way of transforming Boolean into continuous models [33]. More recently, BOOLODE was presented in the context of GRN inference method benchmarking [34, 35], introducing the addition of noise terms to make the ODEs stochastic. By building on top of Boolean networks, these approaches enable to capture the logical and dynamical relationships among the regulators. BOOLODE uses Hill functions to reflect the modulation of gene expression [36–38]. However, this approach relies on a considerable amount of parameters such as mRNA transcription and degradation rates, Hill thresholds and coefficients, signalling timescales, and interaction strengths. Determining these parameters is an important bottleneck as they can hardly be estimated from experimental scRNA-seq data and need therefore to be set arbitrarily or randomly sampled. Moreover, these ODE-based generators fail to produce data with statistical properties comparable to those of real scRNA-seq datasets.

It is crucial that generated count data resemble as much as possible scRNA-seq data to obtain fair inference benchmarks, which implies mimicking dropouts and other statistical features. scBoolSeq relies on the learning of gene-wise RNA pseudocount statistics from a reference dataset. This learning is performed in three steps: (i) the classification of empirical gene pseudocount distributions; (ii) the use of Gaussian Mixtures with up to two components as a parametric model; and (iii) the simulation of dropout events with probabilities that are inversely proportional to the expression value.

scBoolSeq requires the reference dataset to be constituted exclusively of Highly Variable Genes (HVGs). In the literature, HVGs are also referred to as overdispersed genes. This preprocessing step is paramount as it ensures that a coarse-grained view is pertinent. Functions to perform this filtering are available on major scRNA-seq analysis distributions such as STREAM [15] and SCANPY [17]. By selecting HVGs after quality control and normalisation, one ensures that scBoolSeq's reference reflects the underlying biological variation rather than technical noise. In addition to HVGs which are automatically selected by the designated functions in scRNA-seq analysis environments, differentially expressed genes (DEGs) and known

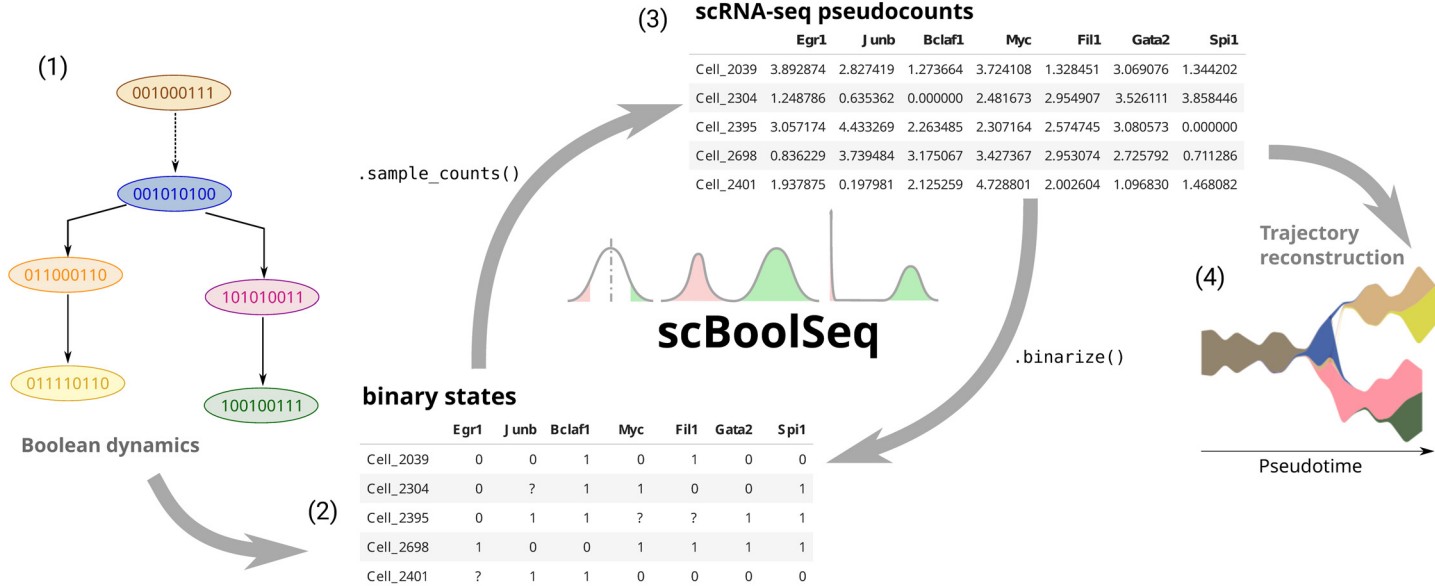

**Fig 1.** From left to right: (1) A branching trajectory constructed by merging two Boolean simulations, each leading to a different stable state. (2) A binarised expression matrix, having genes as columns and samples as rows. (3) A pseudocount matrix (same format as the Boolean matrix). (4) A STREAM-plot reconstructing the branching trajectory from synthetic data generated from the Boolean simulations [15]. scBooLSeq can be used to go from gene expression matrices (such as (3)) to Boolean matrices (such as (2)) and vice-versa.

markers can also be incorporated into scBooLSeq's reference to have a fuller image of the transcriptional landscape of the dynamic phenomenon of interest.

Thus, from the preprocessed reference dataset, scBooLSeq can perform two distinct complementary operations: the binarisation of a scRNA-seq dataset with respect to the reference dataset, and the generation of synthetic scRNA-seq from Boolean activation states, as illustrated by Fig 1.

We first show that our 3-distribution type model of scRNA-seq data and dropouts is able to accurately reproduce the statistical characteristics of a range of scRNA-seq datasets. For the binarisation of scRNA-seq data, we first apply our method to a publicly available scRNA-seq dataset of early retinogenesis. We show that scBooLSeq correctly identifies the different cell types described in the original study, defined by a minimal set of marker genes. These identities can subsequently be used to label cell groups found by the Louvain clustering algorithm [39]. Going beyond cell type identification, we use the Boolean gene activity values determined by scBooLSeq to prune a mouse regulon database [40]. The resulting GRN is validated via Gene Set Enrichment Analysis performed using METASCAPE [41] which yielded numerous relevant Gene Ontology terms related to the kept genes.

Finally, we show that scBooLSeq's synthetic scRNA-seq data generated from Boolean traces produces both discernible trajectories when applying dimensionality reduction techniques and statistics that are comparable to those of real datasets.

Overall, scBooLSeq provides an efficient method to learn statistics of a scRNA-seq dataset and derive binarisation and synthetic generation procedures with few parameters. scBooLSeq has been implemented as an open source Python package available at github.com/bnediction/scBoolSeq.

## Results

### Prerequisites on data preprocessing

In the following, we assume that scRNA-seq data is preprocessed as log pseudocounts $x_{c,g} = log(x_{c,g}^{norm} + 1)$, where $c$ and $g$ refer to cells and genes, respectively. Any size factor-based normalisation can be used, as long as it is of the form $x_{c,g}^{norm} = x_{c,g}^{raw}/S_c$ where $S_c$ is a constant. For instance $S_c = \sum_g x_{c,g}/1e6$ would represent the common library size normalisation, yielding counts per million (CPM). Our methodology is applicable to alternative normalisations such as TPM (transcripts per kilobase per million reads) or RPKM (reads per kilobase per million reads).

Taking the logarithm is necessary to derive reasonable parametric approximations of genewise scRNA-seq distributions: this variance-stabilising transformation mitigates the mean-variance relationship which characterises scRNA-seq data [42]. Furthermore, it reduces the data's skewness [43–46]. Other transformations applicable to scRNA-seq data such as model residuals, inferred latent expression state, and factor analysis present their advantages [46]; but we deem the log to be a good fit for scBoolSeq's purpose: linking scRNA-seq with Boolean dynamics.

In addition to these standard preprocessing techniques, scBoolSeq requires the reference scRNA-seq to be comprised exclusively of Highly Variable Genes (HVGs). The importance of this feature selection step has been highlighted in previous works [47]: Gene Ontology enrichment analysis on HVGs yields terms predominantly related to the phenomenon of interest [47, 48], whereas least-variable variable genes are enriched for translation, mRNA processing, and splicing (housekeeping genes). Diverse software packages for scRNA-seq data analysis such as STREAM [15] Monocle [16], and Seurat [48, 49], to name a few, provide their own version of this procedure. The common idea behind these implementations is decoupling gene expression variation from its mean expression. This is done to select heterogeneous features that will be useful for downstream analyses [47, 49]. In other words, this procedure aims to identify genes whose variability is greater than expected when controlling for the characteristic mean-variance relationship of sequencing data [42, 46]. In our case (linking Boolean dynamics and scRNA-seq data), selecting HVGs favours coarse-graining gene activity when pertinent (i.e., when variability is greater than expected for the given mean expression level, and for genes related to the dynamical phenomenon of interest) and precludes the binarisation of housekeeping genes. Selecting HVGs is out of the scope of this work, nevertheless we provide in S1 Notebooks an end-to-end example of how to perform adequate scRNA-seq data preprocessing using SCANPY (including HVG selection) prior to applying scBoolSeq.

### Classification of Pseudocount distributions and dropout model

scBoolSeq builds on the ideas presented in [50] which seek to capture the different expression patterns across bulk RNA-seq samples of cancer patients. By computing a series of statistical criteria, they proposed to classify empirical pseudocount distributions as bimodal, zero-inflated, or unimodal. This choice of distributions reflects the underlying hypotheses of gene activity: bimodal genes exhibit two distinct expression patterns for the absence and presence of their corresponding encoded proteins. For unimodal genes, we suppose that only cells lying at the tails of the distribution can be confidently inferred to be active or inactive. It also appeared that several genes show a high proportion of zeros, which are then classified as zero-inflated. Their classification method employs statistics such as mean, median, variance, dropout rate, amplitude, Hartigan's dip test for multimodality p-value [51], kurtosis, density peak, and Bimodality Index [52]. As a first step, scBoolSeq discards genes having excessive dropout rates

or failing to exhibit a sufficient dynamic range, by comparing each gene's amplitude to the median amplitude across all genes. In our implementation, the default is to discard genes whose dropout rate is above 0.95, and genes having an amplitude inferior to a tenth of the median amplitude across all genes. This means that they will neither be coarse-grained/binarised nor used to sample synthetic pseudocounts from Boolean gene dynamics. Both the division factor for the median and the dropout threshold are parameters that can be overridden in our Python implementation. Then, bimodal patterns are searched within the kept genes, using a combination of statistics. Next, genes with no bimodal patterns are tested for zero-inflation by looking at the empirical distributions' density peaks. The remaining genes are classified as unimodal.

With scBoolSeq, we generalised and improved this approach (i) to account for the specificities of scRNA-seq data, notably their potential high dropout in gene counts, and (ii) to enable pseudocount sampling from learnt distributions in order to generate synthetic scRNA-seq datasets from Boolean activation states. As we illustrate in S2 Fig, when applied to scRNA-seq, the PROFILE classification scheme and parametric model show two shortcomings: (1) for genes classified as bimodal and unimodal, the dropout tends to artificially decrease their mean and inflate their variance, impeding a good characterisation of their empirical pseudocount distributions via Gaussian or two-component Gaussian Mixtures; (2) for zero-inflated genes, the classification does not result in a parametric distribution, which complicates sampling. We improved the algorithm by computing the statistics on non-zero data and proposed a novel probabilistic dropout model to capture the proportion of zeros. By modelling the probability of a dropout occurring as a function of the expression level with gene-dependent parameters, we were able to reproduce the per-gene dropout rates of different reference datasets. Furthermore, we observed that when sampling from the aforementioned parametric distributions and applying our dropout model, the zero-inflation character of certain genes as well as the excess kurtosis and skewness of unimodal and bimodal genes were globally recovered (S3 Fig).

**Probabilistic simulation of dropout events.** Dropouts arise from both biological (lack of transcription at measurement time) [53] and technical causes (sampling and amplification bias) [54]. For this reason, we built a probabilistic model aiming to: (i) reproduce the distribution of dropout rates across genes in the studied reference datasets; (ii) have a minimal set of gene-dependent parameters; and (iii) have a physical interpretation that accounts for the biological and technical causes of dropouts. Dropout parameters are estimated on a gene-dependent basis because empirical sampling rates exhibit gene-specific bias rather than being uniform random samples of mRNA molecules present in the cell [55]. By modelling these gene-dependent biases and simulating dropout events after sampling from parametric distributions, our dropout method mimics the physical phenomena that give rise to dropout events and generates data that reproduces the statistics of scRNA-seq data, as illustrated by Fig 2.

**Dropout model.** Under the hypothesis that the probability of not observing counts for a certain gene within any given cell is inversely proportional to its relative abundance, the relationship is defined as an exponential decay which has been shown to describe the mean-dropout relationship in several scRNA-seq datasets [56]. We denote by $x_{c,g}$ the prior pseudocount of gene $g$ in cell $c$ and by $x_{c,g}^{\text{obs}}$ the measured pseudocount. The mathematical formulation of the proposed dropout model is of the following form:

$$P(x_{c,g}^{\text{obs}} = 0 \mid x_{c,g}) = \beta_g e^{-\lambda_g x_{c,g}} \tag{1}$$

When simulating dropout events based on these probabilities, the number of dropout events for a given gene across all cells follows a Poisson-binomial distribution [57], that is the discrete probability distribution of a series of independent Bernoulli trials whose success

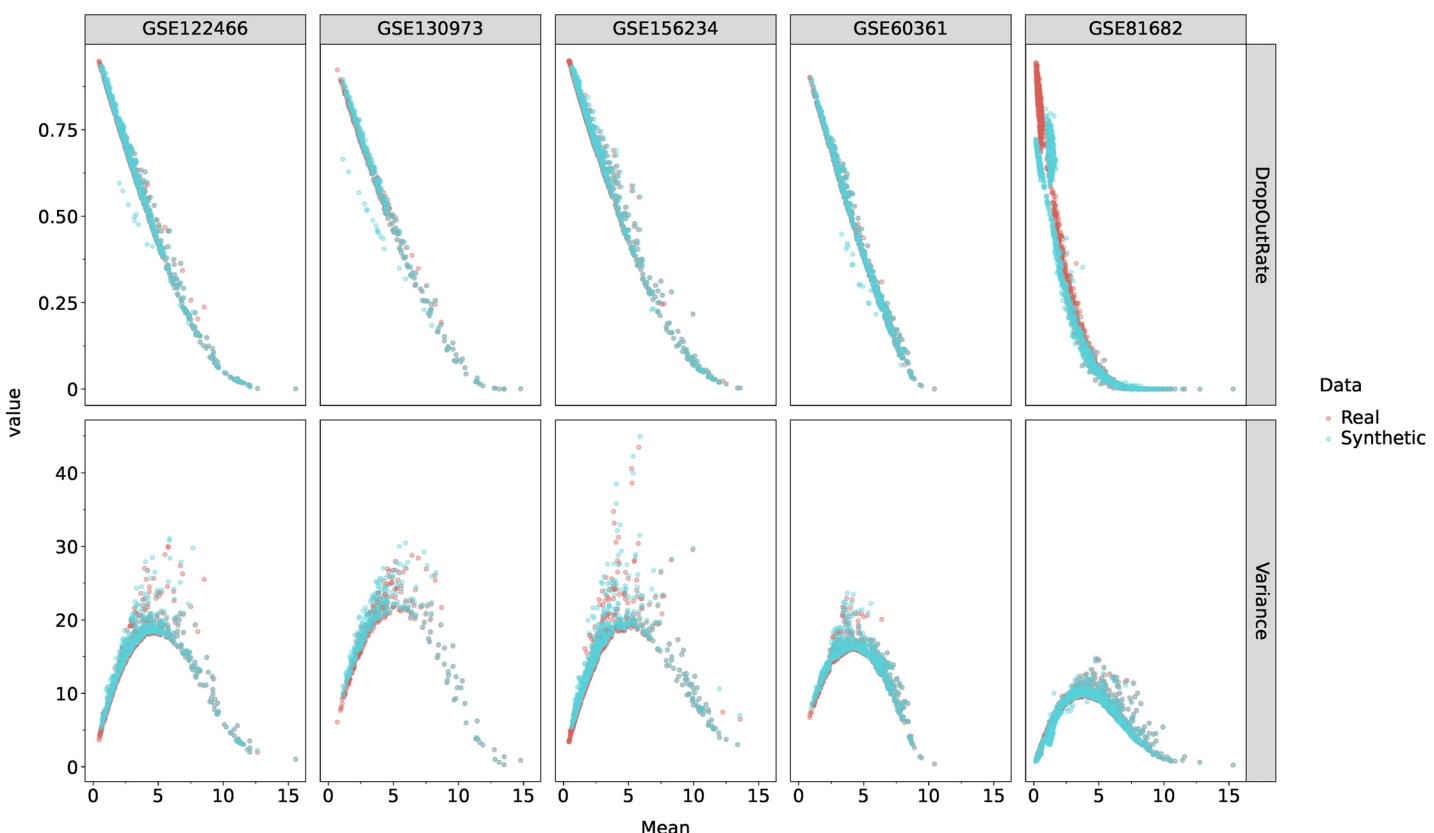

**Fig 2. Mean—Variance, and Mean—Dropout Rate relationships of HVGs in different datasets.** Each blue dot represents the average of 100 samples for a given gene.

(dropout) probabilities are not necessarily identical. This reflects our hypotheses on dropouts: for any given gene, having a dropout event for cell $i$ is independent of the dropout in cell $j$, and two cells having comparable relative transcript abundances of any given gene will have similar probabilities of this gene being observed or dropped out.

**Rate parameter.**    The rate parameter $\lambda_g$ determines the shape of the exponential and thus how rapidly the dropout probabilities decay with the expression value. This parameter is learnt from the reference dataset, independently for each gene, to reflect the aforementioned gene-dependent sampling bias. It is calculated by setting the half-life of Eq 1 to the gene's empirical non-zero mean as follows, for each gene $g$ of the reference dataset:

$$\lambda_g = \frac{ln(2)}{\hat{\mu}_{\text{NZ}}(g)} \tag{2}$$

where $\hat{\mu}_{\text{NZ}}(g)$ is the mean of non-zero pseudocounts of gene $g$ in the reference dataset.

**Normalisation constant.**    The normalisation constant $\beta_g$ is computed from sampled prior pseudocounts as the optimum value minimising the quadratic deviation between the expected dropout rate of the synthetic sample $\text{E}[\tau_g]$ and the reference dropout rate for that gene $\tau_g^{\text{ref}}$ (proportion of zero entries in the reference dataset):

$$\beta_g = \frac{n\tau_g^{\text{ref}}}{\sum_{c=1}^{n} e^{-\lambda_g x_{c,g}}} \tag{3}$$

where $n$ is the number of sampled cells.

This optimum is derived analytically from the expected value of a Poisson-binomial distribution. This ensures that for the same underlying non-zero distribution the dropout rate will, on average, be close to that of the reference.

S1 Fig shows an example of the distribution of rate parameters and the obtained dropout probabilities over the range of expression of a typical log-normalised scRNA-seq dataset. Overall, we observe a trend depending on the gene pseudocount distribution category: for the same sampled value, zero-inflated genes have the highest probability of dropout, followed by bimodal genes. Genes presenting a unimodal distribution have the lowest dropout rates (and highest non-zero means) and therefore will be seldom dropped out.

**Validation.** In order to validate our method, we compared the moments of the experimental scRNA-seq datasets with the moments of data sampled from distributions learnt by scBoolSeq coupled with our dropout model (Eq (1)).

We studied 5 experimental scRNA-seq datasets of the literature on mouse developing retina (GSE122466, [55]); mouse hematopoiesis (GSE81682, [4]); mouse cortex and hippocampus (GSE60361, [58, 59]); human macrophages during efferocytosis (GSE156234; [60]) and ageing human skin (GSE130973, [61]). For each of them, we found that our classification and sampling scheme reproduces extensive statistics of these datasets, especially the gene mean-variance and mean-dropout relationships which characterise scRNA-seq data (Fig 2). Furthermore, the correlation profile between all combinations of mean, variance, skewness, and excess kurtosis is globally recovered (S3 Fig). We find that these correlations are only recovered when applying our dropout simulation method (S9 Fig).

## Binarisation of scRNA-seq data

The coarse-graining scheme of scBoolSeq is based on the classification of pseudocount distributions from a reference dataset, as illustrated by Fig 3. For each gene, cells whose expression

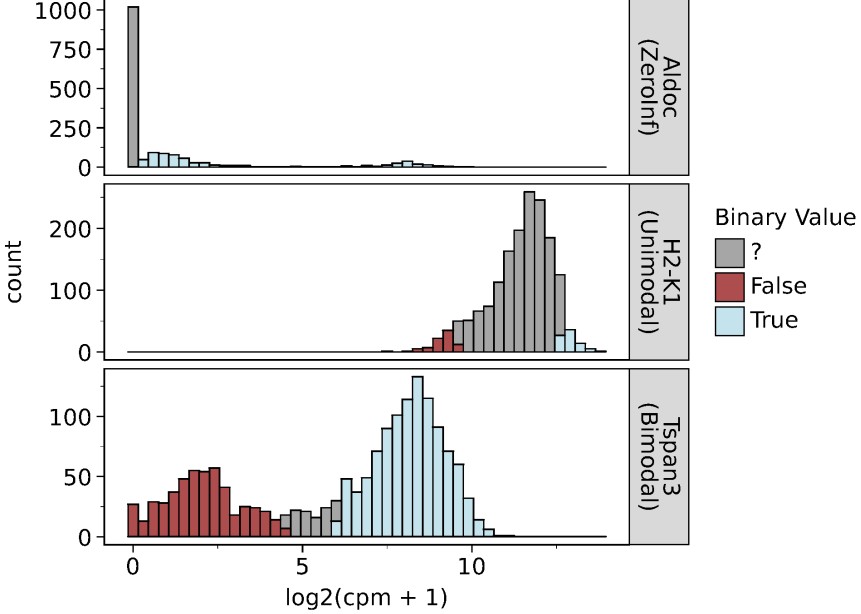

**Fig 3. Illustration of the category-dependent binarisation allows accounting for different shapes in empirical pseudocount distributions.** For each category, plots show the empirical distribution for a selected gene in the GSE81682 dataset, and the part of the values that are binarised with parameters $z$ = "?" for zero-inflated case, $q = 0.05$ and $\alpha = 0$ for unimodal and $\theta = 0.95$ for bimodal.

level is high (respectively low) enough to classify it as True/active (resp. False/inactive) will be binarised whilst cells whose expression level is ambiguous will be left as undefined. As shown in Fig 3, the category-dependent binarisation strategy causes each distribution type to have different proportions of False, True, and undetermined values.

Bimodal genes are binarised using their corresponding univariate two-component Gaussian Mixture Model (GMM), whose parameters are estimated on the reference dataset. The GMM's density is given by Eq 4. The model has two components denoted $C_i$ which are characterised by their parameters $(\phi_i, \mu_i, \sigma_i^2)$. In the following, it always holds that $\mu_2 > \mu_1$, for every bimodal gene. Therefore, we have two components that represent cells whose transcript level can be classified as active $C_2$ or inactive $C_1$.

$$p(x) = \phi_1 \mathcal{N}(x|\mu_1, \sigma_1^2) + \phi_2 \mathcal{N}(x|\mu_2, \sigma_2^2) \;\; s.t. \;\; \phi_1 + \phi_2 = 1 \tag{4}$$

The probabilities of observation $x$ belonging to each one of the two components are first calculated as detailed in Eq (5):

$$p(C_i|x) = \frac{p(C_i)p(x|C_i)}{\sum_{j=1}^{2} p(C_j)p(x|C_j)} = \frac{\phi_i \mathcal{N}(x|\mu_i, \sigma_i^2)}{\sum_{j=1}^{2} \phi_j \mathcal{N}(x|\mu_j, \sigma_j^2)} \tag{5}$$

Then, the binary classification is performed according to a given confidence threshold $\theta$, with $0.5 < \theta \le 1$:

$$b_{\text{bimodal}}(x) = \begin{cases} 0 & \text{if } p(C_1|x) \ge \theta \\ 1 & \text{if } p(C_2|x) \ge \theta \\ ? & \text{otherwise} \end{cases} \tag{6}$$

For genes classified as unimodal, we use thresholds based on two parameters: A *margin* quantile $q$ (0.05 by default) and a multiplier $\alpha$ for the interquartile range IQR. These thresholds are based on Tukey's fences for outlier detection [62, 63], with modified defaults to consistently binarise a small fraction of observations. Note that in Eq (7), $Q(q)$ represents the $q$- th quantile of the gene's empirical distribution. This coarse-graining strategy makes no assumptions about underlying parametric distributions and hence is suitable for binarising genes whose empirical distribution remains skewed after the log transformation. S8 Fig.

$$b_{\text{unimodal}}(x) = \begin{cases} 0 & \text{if } x < Q(q) - \alpha \text{IQR} \\ 1 & \text{if } x > Q(1-q) + \alpha \text{IQR} \\ ? & \text{otherwise} \end{cases} \tag{7}$$

Finally, genes whose empirical pseudocount distribution is classified as zero-inflated use a zero-or-not binarisation scheme [53]. Cells having non-zero counts for a zero-inflated gene are classified as True whilst zero entries can be classified either: (i) as undetermined (parameter $z$ = "?") to reflect the uncertainty regarding their technical/biological causes, or (ii) as False (parameter $z$ = 0), if considered as a signal, as suggested by [53]. Setting the $z$ parameter is a modelling choice, which is easily modifiable in scBoolSeq's Python implementation.

$$b_{\text{zero-inflated}}(x) = \begin{cases} 1 & \text{if } x > 0 \\ z & \text{otherwise} \end{cases} \tag{8}$$

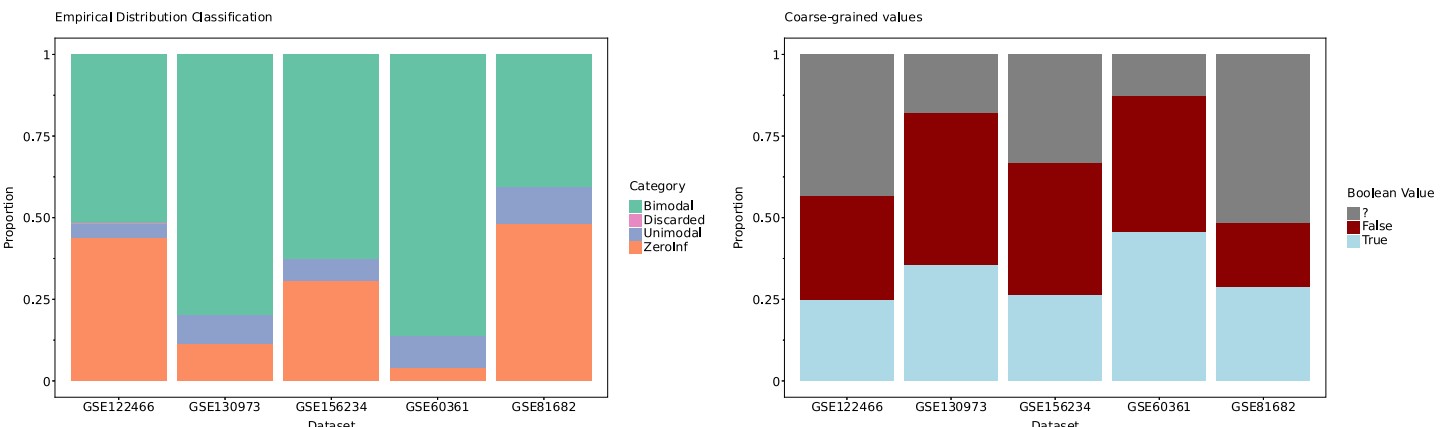

**Fig 4.** Left: Distribution of categories among the studied datasets. Right: Proportion of binarised values across datasets using the default parameters for each distribution type. These proportions are both determined by the categories and the specified thresholds. These were obtained using parameters $z = ?$ for zero-inflated case, $q = 0.05$ and $\alpha = 0$ for unimodal, and $\theta = 0.95$ for bimodal. The dropout rate threshold for marking a gene as discarded was set to 0.99.

The proportion of observations classified as 0 or 1 can be approximated by Eq (9)

$$\xi(1 - \tau) + \beta(p^\star) + \eta(2q) \qquad (9)$$

with the average proportions of binarised observations for each category normalized by the proportion of genes classified as zero-inflated, bimodal, and unimodal, denoted by $\xi$, $\beta$, and $\eta$, respectively, and where $\tau$ represents the average empirical dropout rate.

Fig 4 gives statistics on the fraction of observations that are binarised across the selected evaluation datasets.

In general, zero-inflated genes with a high dropout rate will only have a few observations binarised to 1 and most cells will be classified as undefined. Bimodal genes are binarised across most cells because the underlying Gaussian Mixture correctly describes the bimodal genes' empirical distributions. Finally, unimodal genes will have twice the margin quantile $q$ fraction of observations binarised in the case of $\alpha = 0$ in Eq (7).

**Case study of binarisation: Early-born retinal neurons.** We applied scBoolSeq to a publicly available scRNA-seq dataset to binarise expression data and obtain a qualitative description of phenotypes. We show that the obtained qualitative profiles can serve as a basis to perform inference of Boolean networks, which can mimic the differentiation process and identify key genes and interactions involved in the dynamics.

The dataset originates from [9] (*GEO accession* GSE122466) which analysed how the diversity of cell types found in the early retina (from embryonic days 10 to 17) arises from a pool of progenitor cells. These neurons are retinal ganglion cells (RGCs), cone photoreceptors (cones), horizontal cells (HC) and amacrine cells (AC). The analyses extended previously known marker genes and showed how these appear to be organised in transcriptional waves of co-expression. Extending the original results with a mechanistic model could help formulate hypotheses regarding the underlying regulatory mechanisms of early retinogenesis. Here, we illustrate how to combine the statistical analysis of scBoolSeq to coarse-grain the expression data with prior knowledge data on transcription factor regulations publicly available in the mouse regulon database DoRothEA [40] in order to build logical models that reproduce the differentiation process. Our objective is to first evaluate how the binarisation preserves the cell type classification, and how the resulting qualitative description of phenotypes enables to identify core regulations that explain the Boolean differentiation process.

**Discriminating cellular types using prior-knowledge markers.** The reference study [9] considered prior knowledge markers for the cellular types at different stages of differentiation. We classified each cell according to its binarised expression profile and the markers it contains. Then, for each cellular type, we computed how many cells have the matching marker, and among them, how many match only with that cellular type. As shown in Table 1, the majority of cells per group were unambiguously identified, except for Horizontal Cells. Notice that Horizontal Cells share one marker *Prox1* with Amacrine Cells. It should be noted, that in this case, a quarter of all considered cells have been classified using their binarisation (S4 Fig). Moreover, our classification of cells based on their binarised pseudocounts and prior-knowledge markers enables to label Louvain clusters of scRNA-seq data, which turned out to be consistent with labels obtained using differential expression analysis by [9] (S5 Fig).

**Data-driven inference of Boolean models.** The binarisation of scBoolSeq enables to specify Boolean dynamical properties that reflect the observed differentiation process: the existence of traces linking (partially) binarised cellular states, including branches from pluripotent states to distinct differentiated states, as well as stability properties. Then, inference methods such as BoNesis [64, 65] can derive Boolean networks that reproduce the specified dynamics. The logical rules are derived from prior knowledge Gene Regulatory Networks (GRNs), typically extracted from TF-TF (transcription factor—transcription factor) interaction databases, possibly completed with statistical network inference from scRNA-seq data. By employing combinatorial optimisation method, BoNesis enables accessing to the sparsest models, i.e., requiring as few as possible genes to reproduce the desired traces and stable states.

Using clustering and trajectory reconstruction methods, we applied scBoolSeq to determine a partial binary profile of 6 cellular types, namely RPC (progenitor), intermediate neuroblast types NB1 and NB2, and final Cones, RGC and AC types. Note that due to the low number of cells classified as HC and their apparent distance between each other, we omitted this cellular state. The dynamical specification consisted of the existence of a traces from the RPC state to NB1 and then to NB2. From the NB2 state, three different traces must exist towards each of the final stable states. Moreover, we extracted from the DoRothEA database a core TF-TF regulatory network together with target genes that have been binarised. Focusing on the largest weakly connected component, it gave a GRN with 644 genes. Then, using BoNesis, we reconstructed Boolean networks that, using the input GRN interactions, can reproduce the desired traces and stable states. See Methods section and S7 Fig for details. Because the binary profiles are partial, numerous genes have no imposed binary value in several cellular states. Using BoNesis, we identified models that rely as little as possible on the dynamics of those genes with undetermined states. It resulted in pruning the input GRN to 184 genes which suffice to explain the observed differentiation process. As shown in Fig 5(Right), gene

**Table 1. A list of all the cellular types of interest, as well as the Boolean markers (cells with those genes binarised to 1/True/active) used to detect cells matching belonging to them.** *N. Unambiguous Cells* represents cells that exclusively expressed the given set of markers.

| Cell Type | N. Unambiguous Cells | Perc. Total | Markers |
|---|---|---|---|
| RPC (Retinal Progenitor Cells) | 249 | 98.03% | Sox2, Fos, Hes1 |
| NB1 (Neuroblasts, first group) | 23 | 85.19% | Top2a, Prc1, Sstr2, Penk, Btg2 |
| NB2 (Neuroblasts, second group) | 27 | 81.82% | Neurod4, Pax6, Pcdh17 |
| RGC (Retinal Ganglion Cells) | 191 | 94.55% | Isl1, Pou4f2, Pou6f2, Elavl4 |
| AC (Amacrine Cells) | 81 | 67.50% | Onecut2, Prox1 |
| HC (Horizontal Cells) | 3 | 10.71% | Onecut1, Prox1 |
| Cones (Photoreceptors) | 8 | 100% | Otx2, Crx, Thrb, Rbp4 |

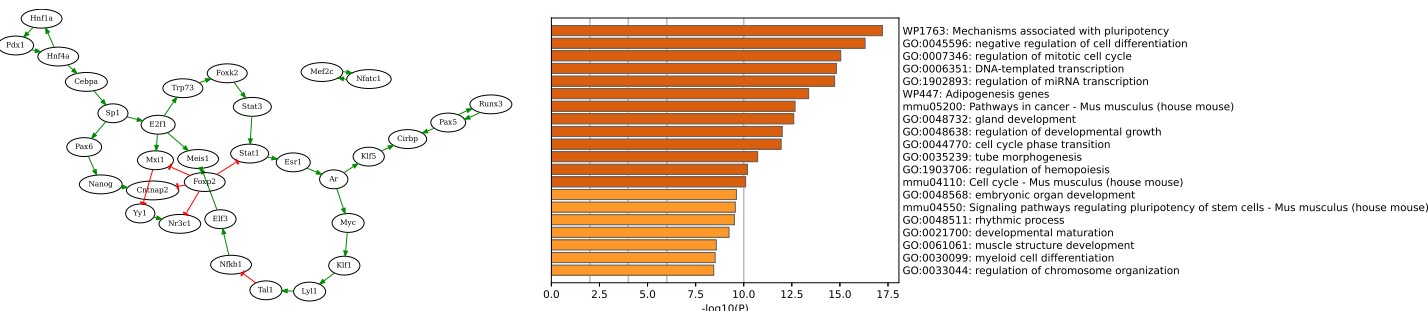

**Fig 5. Left**: Simplified view of the set of minimal TF-TF interactions employed in the Boolean models reproducing the differentiation process. For display, all leaf nodes with an in-degree of 1 were recursively removed from the GRN. The full filtered GRN obtained with BoNesis is provided in S6 Fig. **Right**: Top Gene Ontology Terms related to the 184 genes of the filtered GRN.

ontology enrichment analysis, performed using Metascape [41], shows many relevant ontology terms were found among the top hits, such as mechanisms associated with pluripotency, negative regulation of cell differentiation, regulation of mitotic cell cycle, gland development, regulation of developmental growth, and embryonic organ development. Obtained models can then serve as inputs for a more thorough systems biology analysis of the biological question.

## Synthetic scRNA-seq generation biased by Boolean states

As the inverse operation of binarisation, the parametric distributions and dropout model learnt per gene from a reference dataset also enable the generation of synthetic pseudocounts from Boolean activation states. This can be achieved by biased sampling from distributions learnt on non-zero entries of the reference dataset. Consequently, dropout events are simulated according to the gene-dependent model of Eq (1).

Biased sampling ensures that cells in which a gene is active will exhibit higher expression (pseudocounts) than those in which it is inactive. In the case the gene follows a unimodal distribution of mean $\mu$ and variance $\sigma^2$, the pseudocount are sampled from the half-normal distribution corresponding to the activation state ($\mathcal{HN}(\mu, \sigma^2)$ for active, and $\mu - \mathcal{HN}(0, \sigma^2)$ for inactive). In the case of bimodal distribution, composed of two normal distributions of mean $\mu_1 < \mu_2$ and variance $\sigma_1$ and $\sigma_2$, respectively, the sampling is performed from the mode corresponding to the activation state ($\mathcal{N}(\mu_1, \sigma_1^2)$ for inactive and $\mathcal{N}(\mu_2, \sigma_2^2)$ for active). Finally, in the case of zero-inflated genes, learning from non-zero entries ensures falling back to one of the two aforementioned cases, and the dropout model learnt should reflect the inflation of zeros. The last step simulates dropouts in such a way that synthetic log-pseudocounts produced from the Boolean states will have gene-wise statistical properties closely resembling those of real scRNA-seq data. The dropout event simulation can follow the dropout model of Eq (1) learnt per gene, or follow an arbitrary given distribution.

**Application to artificial Boolean models.** The above steps enable the generation of synthetic scRNA-seq datasets from collections of binary states of genes, as it would be typically generated from the simulation of Boolean networks [66, 67]. These synthetic data can then serve as a basis for benchmarking inference methods, with known ground-truth dynamical model. This could notably be applied to artificial Boolean models of different scales and topology. In this case, however, nodes are not directly referring to the genes of an experimental scRNA-seq reference dataset.

A possible approach, proposed in scBoolSeq, is to analyse the shape of the node-wise distribution of Boolean values and assign genes having similar shape. Intuitively, a gene is for

instance active in most cells, it can be classified as unimodal. Subsequently, genes that vary considerably can be considered to be bimodal. Genes that are ubiquitously inactive with few exceptions (e.g., it is active in only one state of the Boolean trace) would then be zero-inflated. scBoolSeq uses scaled versions of the first four moments to classify Boolean gene distributions as unimodal, bimodal, or zero-inflated. The scaled moments of Boolean distributions are fed to a k-nearest-neighbours classifier [68, 69] that was trained on the scaled moments of the reference dataset, using their corresponding distribution types as labels. Next, for each category, a unique one-to-one mapping is performed between Boolean nodes and genes in the reference scRNA-seq dataset. Therefore, each Boolean node will be represented by one gene in the reference which cannot be assigned to any other Boolean node. This allows the generation of synthetic scRNA-seq data whose gene-wise distributions correctly represent the underlying Boolean dynamics.

We applied this principle to three artificial Boolean models, exhibiting different types of emerging dynamics. For each of the models, Boolean traces representing the dynamics of the network were obtained as described below. For each of the Boolean states of the traces, multiple synthetic pseudocounts were sampled using scBoolSeq with the selected reference dataset GSE81682. Then, we applied classical scRNA-seq dimensionality reduction methods to visualise the corresponding pseudocount trajectories. Further details regarding the sampling procedure and projections can be found in S1 Notebooks.

The first artificial model is a star-like network (Fig 6a) in which a single transcription factor (TF) up-regulates the expression of a set of genes. This model was simulated by performing one random walk with the fully asynchronous update mode starting from the state where the node tf is active and all genes are inactive. The resulting trace is a sequence of Boolean vectors where genes progressively activate, in a random order. This gradual activation can be clearly distinguished in Fig 6b, where cells with few active genes are coloured in dark blue and cells with all genes active are coloured in light green.

The second manually-designed model is a bistable switch which represents a simplified *cellular reprogramming* scenario (Fig 6c) in which the cell finds itself in a steady state (light blue, labelled *common*) characterised by the activation of *TF6* which activates a small set of genes and inhibits a mutually exclusive switch. The activation of *TF7* node represents a perturbation that inhibits *TF6*, pushing the cell out of its initial state and triggering a differentiation process. One of two different stable states is eventually reached.

The third model (Fig 6) is a three-stable switch that has been designed automatically from random scale-free topology such that it exhibits a two-level differentiation process: from an initial state, three stable states are reachable, with an intermediate branching state giving access to two of them. In both cases, we generated Boolean traces covering the differentiation branches from the initial states. These traces remain apparent in the projections of generated scRNA-seq data (Fig 6d and 6f).

**Comparison with BoolODE.** Given an artificial Boolean network, the tool BoolODE [34] is capable of producing synthetic pseudocount datasets which exhibit clearly defined trajectories when applying dimensionality reduction techniques such as *t-SNE*. However, datasets generated with BoolODE exhibit statistics that do not resemble those of scRNA-seq data.

Fig 7 provides comparisons between datasets generated by BoolODE and scBoolSeq from one of the largest curated models of the benchmark presented in [34], a Boolean network of human gonadal sex determination (GSD) [70]. It has two main fixed point attractors of biological interest, namely Sertoli cells and granulosa cells which correspond to male and female phenotypes. We notably compared the mean-variance and mean-dropout profiles of generated data with different dropout models, as proposed by both tools. Besides the dropout rate being constant, the mean-variance relationship of BoolODE appears to be at a very different scale

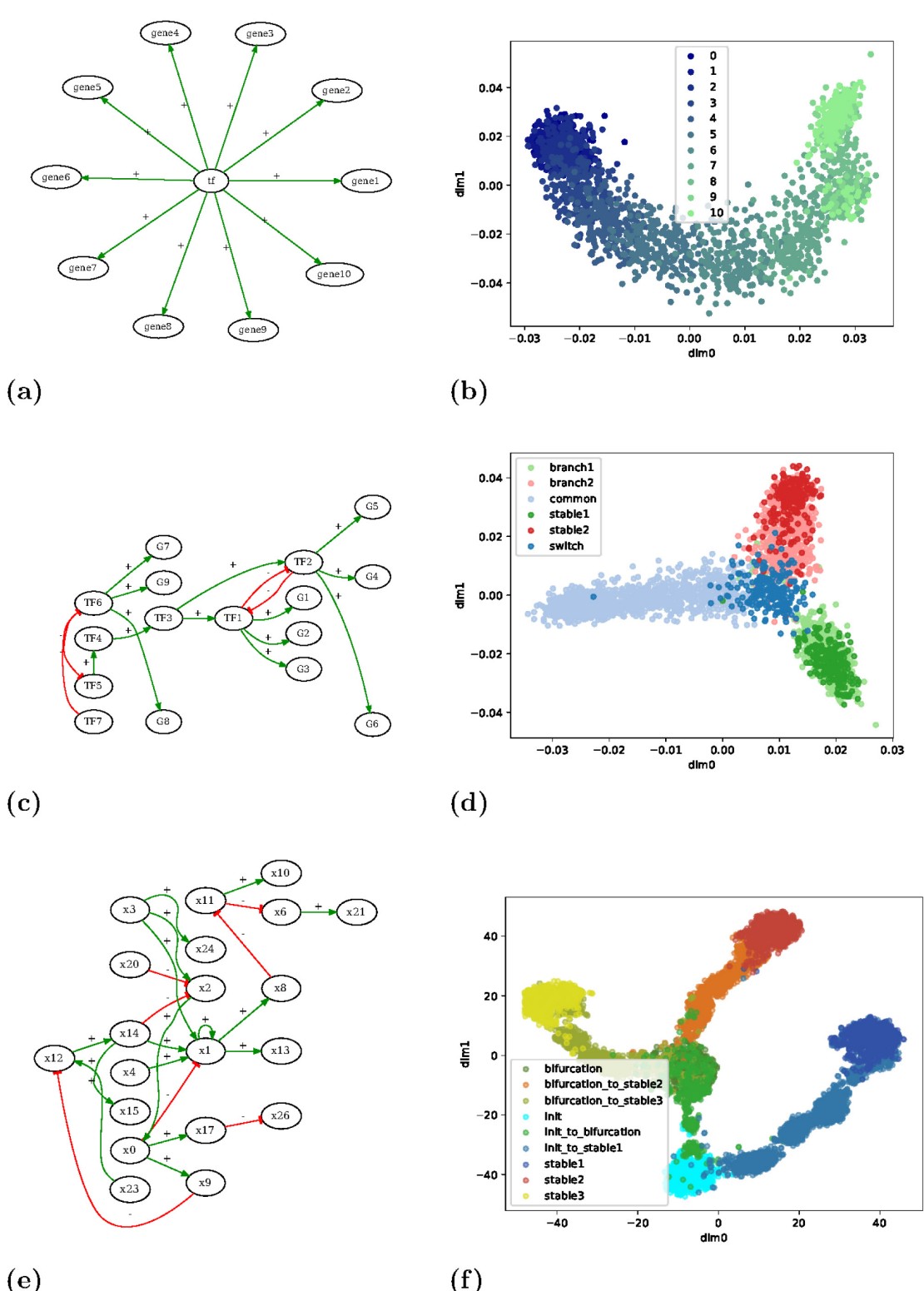

**Fig 6. Artificial Boolean models and generated synthetic scRNA-seq data. Left**: Influence graphs of the Boolean models. See S1 Notebooks for Boolean functions. **Right**: Two-dimensional projection of the synthetic scRNA-seq data generated by applying scBoolSeq to Boolean traces simulated from the models on the left; we used PCA and locally linear embedding (LLE) for (b) and (d), and t-SNE for (f). Dots are labelled with a description of the Boolean state they have been generated from: for (b) it is the number of active genes; for (d) and (f) they refer to the dynamical nature of the states in the 3-branches of the differentiation process.

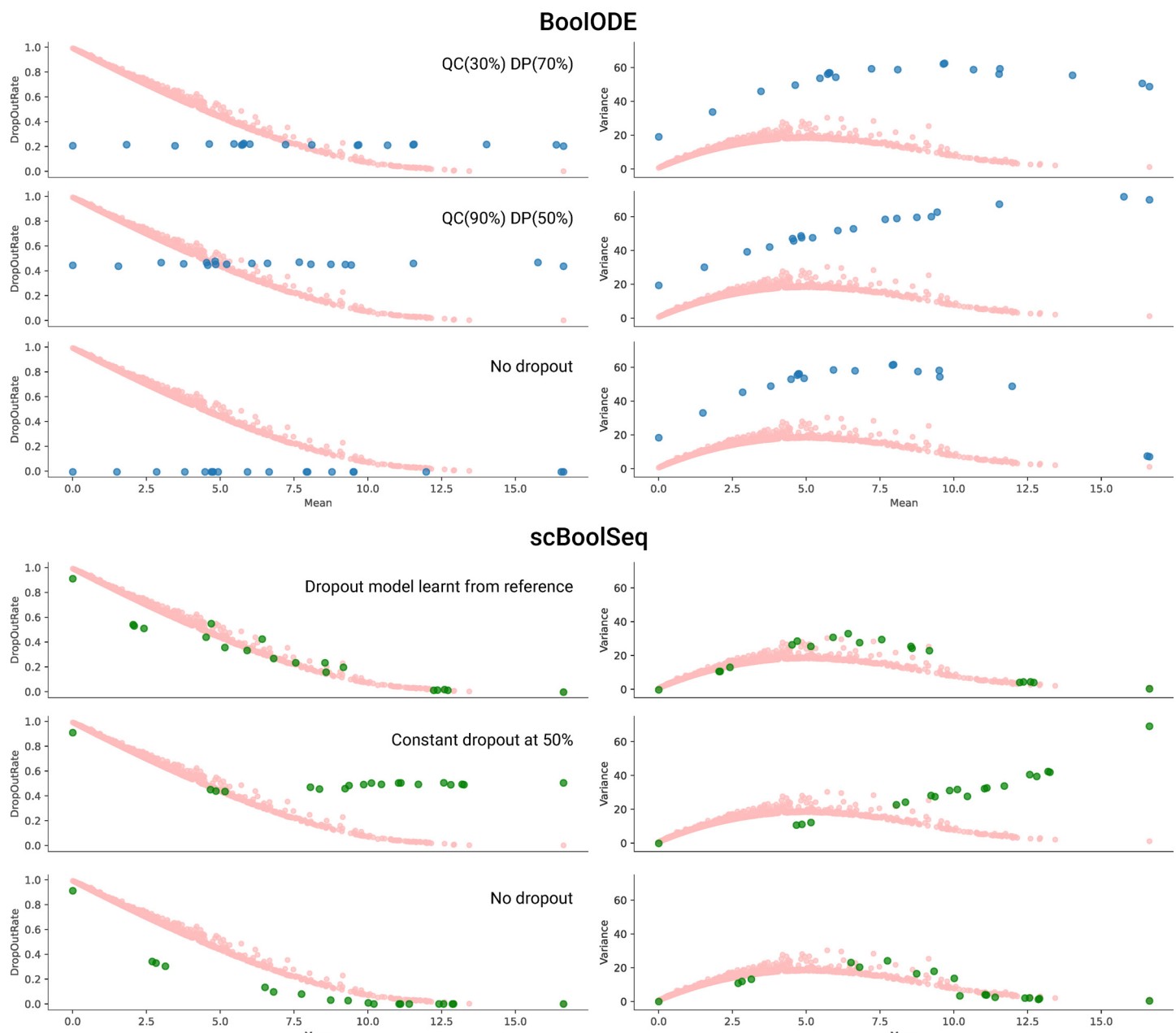

**Fig 7. Comparison of the per-gene Mean-Variance and Mean-DropOutRate profiles of reference dataset GSE122466 (red), BoolODE (blue), and scBoolSeq (green).** *QC* represents the quantile below which BoolODE simulates dropouts with a constant probability *DP*.

than typical scRNA-seq data (Fig 2). It should be noted that when enforcing a constant dropout rate with scBoolSeq, the resulting dropout-mean profile is not constant as 0 values can still be sampled from learnt pseudocount distributions: Gaussian distribution can give non-zero probabilities to negative values, which are corrected as 0. This is not the case with boolODE because of the noise added to ODE-simulated values, which prevents generating values being exactly 0.

```python
import pandas as pd
from scboolseq import scBoolSeq

# cells are rows and genes are columns
reference = pd.read_csv("reference_scRNA_highly_variable_genes_pseudocounts.csv")

scbool = scBoolSeq()
# compute criteria (statistics and per-gene category)
scbool.fit(reference)

# binarise the reference dataset (or other)
coarse_grained = scbool.binarize(reference)

# Simulate scRNA_Seq experiments from Boolean data
boolean_states = pd.read_csv("simulated_boolean_dynamics.csv")
synthetic_rna = scbool.sample_counts(boolean_trace)
```

**Fig 8. Python code snippet showing basic usage of scBoolSeq for binarisation and synthetic data generation from reference scRNA-seq data and Boolean states.**

## Implementation and usage

scBoolSeq has been implemented in Python on top of `pandas` [71], `statsmodels` [72], and `scikit-learn` [68] libraries. Fig 8 shows basic usage of `scBoolSeq` to perform binarisation and synthetic data generation. In terms of scalability, scBoolSeq's current implementation requires loading the full pseudocount matrices in memory. In the case of these 5 datasets, their size ranges from 3,000 to 15,000 single-cell measurements over 300 to 1,200 genes. In each case, the learning could be handled with less than 1GB of RAM and took less than a minute on a 3Ghz CPU with 16 threads. The total binarisation and sampling operations ran in seconds. Future engineering work will focus on leveraging the `AnnData` [73] Python package for handling large datasets that cannot be fit in RAM. Furthermore, using `AnnData` within scBoolSeq will allow its integration in the `scverse` [74] computational ecosystem for single-cell omics data analysis.

scBoolSeq is distributed as a standard Python package and is integrated into the CoLo-MoTo Docker distribution [75], which facilitates the accessibility of tools related to Boolean and logical models, and the reproducibility of related computational analyses.

## Discussion

We introduced scBoolSeq, a novel method that provides a bidirectional link between scRNA-seq data and Boolean Models. Our method builds on the classification of gene empirical pseudocount distributions into unimodal and bimodal distributions proposed by [50], that we extended with a probabilistic gene-dependent dropout model. We showed that the resulting characterisation suffices to capture the main statistical features of real scRNA-seq data. Then, scBoolSeq offers both the ability to binarise scRNA-seq datasets and the ability to generate synthetic pseudocounts from binary states of genes.

### From pseudocounts to binary states

We illustrated on a concrete application how the binarisation offered by scBoolSeq can be employed to process scRNA-seq data in view of performing inference of Boolean networks, which are logical models of gene activity dynamics. First, scBoolSeq coarse-graining method allows identifying cellular types of interest by detecting the presence (i.e., activation) of known

marker genes. In addition to this, combining the binarised gene activity with community detection techniques could help to find previously unknown marker genes (genes that are binarised as active only in certain clusters and are not found in the literature). Then, coupled with a prior GRN, the deduced set of Boolean functions constitute a set of hypotheses that can guide future wet-lab experiments to unveil the core regulatory mechanism of early retinogenesis.

It should be stressed that the binarisation of scBoolSeq can result in an undetermined state when there is not enough statistical evidence for a binary classification. We believe that the fact that not all genes (and cells) can be classified with binarisation is a biologically meaningful feature as the method enables discriminating cells in extreme (asymptotically stable) states from cells in transient states, for which a fully binary view may not be adequate.

One should note however that determining the activity of a gene based on its transcript level is a strong hypothesis. Methods such as VIPER [76] aim at adding information about each protein's regulon to better infer protein activity. Moreover, chromatin accessibility data and other epigenetic information can also help to refine the binary classification.

## From binary states to pseudocounts

Another major contribution of scBoolSeq is its method for generating synthetic scRNA-seq data from Boolean gene activation states by biased sampling from learnt pseudocount distributions on a reference dataset. We showed that scBoolSeq provides a significant improvement over BoolODE as it produces synthetic scRNA-seq data whose statistical characteristics (mean-variance and mean-dropout profiles) closely resemble those of real data. In addition to this, scBoolSeq allows the simulation of any arbitrary distribution of gene-wise dropout rates. This represents an unprecedented contribution as it allows measuring the sensitivity of inference methods to the dropout rate distributions of scRNA-seq datasets.

By offering the capability to generate synthetic scRNA-seq datasets from ground-truth Boolean models with realistic statistical features, we believe that scBoolSeq is a clear asset for that generating benchmarks for the evaluation of various inference methods, such as GRN inference, trajectory reconstruction, and data-driven Boolean network inference.

## Methods

### Boolean networks and dynamics

A *Boolean network* on nodes $\{1, \ldots, n\}$ is a function $f : \mathbb{B}^n \to \mathbb{B}^n$ mapping binary vectors of dimension $n$ to themselves, where $\mathbb{B} = \{0, 1\}$ is the Boolean domain. For each node $i \in \{1, \ldots, n\}$, we write $f_i : \mathbb{B}^n \to \mathbb{B}$ the $i$-th component of $f$, which is the Boolean function of node $i$. A Boolean vector $\mathbf{x} \in \mathbb{B}^n$ specifies a Boolean state for each component of the network, and is called a *configuration*.

The *influence graph* of a Boolean network $f$ is a directed signed graph, noted $G(f)$, whose vertices are the nodes of the Boolean network. The influence graph captures the dependencies of Boolean functions, and corresponds to union of Jacobian matrices of $f$ on configuration. Formally, there is a positive edge for node $j$ to $i$ ($j \xrightarrow{+} i \in G(f)$) in the influence graph if and only if there exists a configuration $\mathbf{x} \in \mathbb{B}^n$ such that

$$f_i(\mathbf{x}_1, \ldots, \mathbf{x}_{j-1}, 0, \mathbf{x}_{j+1}, \ldots, \mathbf{x}_n) < f_i(\mathbf{x}_1, \ldots, \mathbf{x}_{j-1}, 1, \mathbf{x}_{j+1}, \ldots, \mathbf{x}_n)$$

There is a negative edge for node $j$ to $i$ ($j \xrightarrow{-} i \in G(f)$) in the influence graph if and only if there exists a configuration $\mathbf{x} \in \mathbb{B}^n$ such that

$$f_i(\mathbf{x}_1, \ldots, \mathbf{x}_{j-1}, 0, \mathbf{x}_{j+1}, \ldots, \mathbf{x}_n) > f_i(\mathbf{x}_1, \ldots, \mathbf{x}_{j-1}, 1, \mathbf{x}_{j+1}, \ldots, \mathbf{x}_n)$$

Note that it is possible to have both edges $j \xrightarrow{+} i$ and $j \xrightarrow{-} i$ in a same influence graph. If it is the case for $G(f)$, then the Boolean network $f$ is said to be non-monotone. Otherwise, $f$ is *locally monotone*.

A *trace* of a Boolean network $f$ is a sequence of configurations $\mathbf{x}^1, \cdots, \mathbf{x}^k$ that can be computed according to a given *update mode*. For instance, the synchronous mode computes traces such that any two successive configurations $\mathbf{x}^m, \mathbf{x}^{m+1}$ are such that $\mathbf{x}^{m+1} = f(\mathbf{x}^m)$; the fully asynchronous update mode computes traces such that any two successive configurations $\mathbf{x}^m, \mathbf{x}^{m+1}$ differ on only one node $i$, and verify that $\mathbf{x}_i^{m+1} = f_i(\mathbf{x}^m)$. The *most permissive* update mode [24] computes all the traces that are binarised from any asynchronous traces of multivalued and quantitative model compatible with the Boolean network. In general, it allows much more traces than synchronous and (general) asynchronous modes, which fail to capture traces of different classes of quantitative systems, including incoherent feed forward loops [24].

A configuration $\mathbf{x} \in \mathbb{B}^n$ is a *stable state* if $f(\mathbf{x}) = \mathbf{x}$, i.e., it is a fixed point of $f$. A configuration $\mathbf{x} \in \mathbb{B}^n$ belong to an *attractor* of $f$ under a given update mode whenever for any possible traces from $\mathbf{x}$ to another configuration $\mathbf{y}$, there exists a traces going back to $\mathbf{x}$. Stable states are particular cases of attractors.

## Inference of Boolean networks from influence graph and dynamical properties

From an influence graph $\mathcal{G}$ and a set of dynamical properties, the tool BoNesis [64, 65] (bnediction.github.io/bonesis) allows inferring Boolean networks $f$ having their influence graph enclosed by $\mathcal{G}$, i.e., $G(f) \subseteq \mathcal{G}$, and that posses the input dynamical properties. The dynamical properties supported by BoNesis include the existence of most permissive traces between partially specified configurations, and stable state properties of (partially specified) configurations. A partially specified configuration specifies a Boolean state for a subset of nodes. The states of the other nodes can then be freely determined in order to satisfy the dynamical properties. BoNesis also allows specifying optimisation objectives to filter solutions, notably to enumerate only sparser models, i.e., with the smallest influence graphs. The Boolean networks returned by BoNesis can then be exported to standard textual representation for further analysis with other software tools.

We employed BoNesis to infer Boolean networks from scRNA-seq scBoolSeq binarisation (see next section), and to generate artificial Boolean networks which possess multi-stability and branching behaviours from randomly generated scale-free influence graph (S1 Notebooks).

## Case study: Early born retinal neurons

We performed the analyses on the scRNA-seq dataset of lane 1 of GSE122466. The main steps hereafter denoted in paragraphs refer to the analyses performed in their homonymous Jupyter Notebooks provided in S1 Notebooks.

**Highly variable gene selection.** For this part we used the software STREAM [15]. We took the count matrix of the first replicate (Identified with the prefix `Lane_1` in their index). We performed standard quality control, with the same parameters as the analyses of the original article. Cells expressing less than 200 genes were discarded, as well as genes expressed in less than 3 cells. We selected the 1648 most highly variable genes and appended to them the two marker genes that were reported in the article but were not selected as being highly variable (*Rbp4*, *Pou4f2*).

**Retinal differentiation clustering and metadata.** In this part we took the aforementioned Highly Variable Genes (HVGs) and performed the scBoolSeq distribution learning with

$\theta$ = 0.75 to have a higher amount of binarised observations on bimodal genes. We then used the instance to binarise the HVGs across all cells. We then identified cells matching the markers described in the original article. About 25% of all cells were labelled in this process. Subsequently, cells matching more than one set of markers were discarded. The only pair of phenotypes which presented more than a couple of ambiguous cells were Amacrine Cells (AC) and Horizontal Cells (HC) which had 23 cells matching both marker signatures. This was expected given that cellular types were defined with only two markers and one of them *Prox1* is shared. Having a larger (and preferably disjoint) set of markers could resolve this ambiguity. We used SCANPY [17] to perform Louvain clustering on the log pseudocount HVGs, with the number of neighbours set to 15. With this analysis, 11 distinct clusters were found. A small cluster of cells (cluster 10 in the notebooks) was discarded as it was determined to be an unknown cluster of unknown Retinal Ganglion Cell-like U/RGC. Our Boolean analysis also found this isolated cluster to express signature genes of RGCs. Finally, clusters were labelled using the majority label of cells whose Boolean identity matched the markers. Most clusters had absolute majorities (85%, 98%) except for one (Cluster 3 had 53.84% of cells voting NB2, and 34.61% voting AC: It was labelled NB2). These labels were used as metadata to perform trajectory inference.

**Trajectory inference.** Using STREAM we performed trajectory inference, using the aforementioned cluster labels as metadata. We obtained a well-defined trifurcating trajectory that is distinguishable on two dimensions. We set the root (starting point) to be Retinal Progenitor Cells (RPCs) and the three final points to be the Cones, Retinal Ganglion Cells (RGC), and Amacrine Cells respectively. Cells associated with these terminal nodes were taken as representative of their corresponding phenotypes. For the two groups of neuroblasts (NB1 and NB2), cells within the two quartiles $Q(.25)$, $Q(.75)$ of the root node's pseudotime were chosen as representative of these transient phenotypes. This yields a total of 133 RGC, 79 NB1, 17 NB2, 109 AC, 78 RPC, and 69 Cones that were used to infer the Boolean model.

**Binarisation of scRNA-seq data.** We binarised all HVGs across all cells and employed the metadata obtained from the previous trajectory inference step to retrieve cell groups. We defined meta-observations by aggregating each group, using the mode as summary statistic. We further selected genes having a non-null variance, which reduced the original 1650 genes to only 1426. We only retained binarised genes present in the mouse regulon database DoROTHEA [40], that is 1263.

**Boolean model inference.** Having our binarised observations and selected genes, we defined our GRN using DoROTHEA [40]. DoROTHEA gives a confidence score to each one of the interactions, based on the number of supporting evidence in different sources. In decreasing order, these levels are: *A,B,C,D,E*. We decided to exclude interactions with low supporting evidence, so we filtered out levels *D,E* and considered only levels *A,B,C*. With these filtered interactions, we extracted the core TF-TF network which we define to be the biggest strongly connected component of the departing graph. This core TF-TF network has 157 nodes. We then obtained the subgraph induced by these 157 core transcription factors and the binary genes comprising our observations. This yielded a GRN with 728 nodes. We tested and found that this GRN was not weakly connected. We extracted the biggest weakly connected component which contained 633 nodes. This weakly connected component was given to BoNesis as the domain of Boolean Networks to consider, and specified the desired traces and stable states using the specification given in S7 Fig.

## Supporting information

**S1 Notebooks. Notebooks and Boolean networks for reproducing binarisation case study and synthetic data generation.** The notebooks are provided as static HTML files, and Boolean

networks as textual files in BoolNet format. See the Data availability statement for links to executable notebooks and code.
(ZIP)

**S1 Fig. Example of distribution of rate parameters and dropout probabilities learnt by scBoolSeq.** Left: Distribution of rate parameters λ estimated on dataset GSE122466. Right: Dropout probabilities computed between the minimum and maximum values of a sample from the parametric distributions corresponding to the same dataset. Each line corresponds to an individual gene.
(PDF)

**S2 Fig. Mean—Variance and Mean—DropOutRate relationships of HVGs using PROFILE parametric distibutions for bimodal and unimodel genes on selected scRNA-seq datasets.** Each green point represents the average of 100 independent replicates with the same sample size as the reference dataset.
(PDF)

**S3 Fig. Correlation between higher moments of real pseudocount data and from data generated from distributions and dropout model learnt by scBoolSeq on selected scRNA-seq datasets.**
(PDF)

**S4 Fig. Position of cells classified using scBoolSeq binarisation and prior-knowledge markers.** t-SNE and UMAP projections trained on the top 25 principal components (log pseudocount matrix). Colours indicate cell identities determined by binary value of known markers (see Table 1).
(PDF)

**S5 Fig. Result of trajectory reconstruction using STREAM on early-born retinal neurons scRNA-seq data.** UMAP projection of the first 25 principal components to 3 dimensions (only 2 are shown). The cluster labels are determined by the majority label of unambiguous cell types identified via scBoolSeq binarisation.
(PDF)

**S6 Fig. Influence graph of sparser Boolean networks learnt using BoNesis from qualitative dynamics of case study obtained with scBoolSeq binarisation.** This graph comprises 184 nodes forming Boolean networks that can reproduce the Boolean dynamics of early-born retinal neurons differentiation process. This graph is a subgraph of the input DoRothEA TF-TF interaction database. Green arrows indicate positive regulations, red arrows indicate negative regulations. Nodes without predecessors indicate nodes with constant function in the Boolean networks. Thus, the Boolean state of these nodes is identical in all stable states, and is in opposite state in the precursor state RPC.
(PDF)

**S7 Fig. Python code snippet showing usage of BoNesis for the inference of Boolean networks for the retinal differentiation case study.** See S1 Notebooks for full pipeline.
(PDF)

**S8 Fig. Disambiguation of Unimodal genes' coarse-graining and sampling parametrisations.** Genes classified as Unimodal may exhibit heavy tails or remain skewed after the preprocessing log-transformation step. However, these characteristics do not hinder their coarse-graining: scBoolSeq uses a nonparametric quantile-based binarisation scheme that makes no assumptions about the underlying distribution. This is independent from the biased sampling

procedure: By using half-normal distributions, scBoolSeq produces synthetic data reflecting unimodal activation patterns found in Boolean gene dynamics.
(PDF)

**S9 Fig. Importance of scBoolSeq's Dropout model.** A Gaussian distribution (for Unimodal Genes) or two-component Gaussian Mixture (for Bimodal Genes) by themselves do not suffice to capture the statistical characteristics of log-transformed and normalised Highly Variable Genes of scRNA-seq datasets. However, when combined with our probabilistic dropout model these parametric distributions are able to recover the statistics of these data.
(PDF)

## Author Contributions

**Conceptualization:** Gustavo Magaña-López, Laurence Calzone, Andrei Zinovyev, Loïc Paulevé.

**Data curation:** Gustavo Magaña-López.

**Funding acquisition:** Loïc Paulevé.

**Methodology:** Gustavo Magaña-López, Laurence Calzone, Andrei Zinovyev, Loïc Paulevé.

**Software:** Gustavo Magaña-López.

**Supervision:** Andrei Zinovyev, Loïc Paulevé.

**Validation:** Gustavo Magaña-López, Loïc Paulevé.

**Visualization:** Gustavo Magaña-López, Loïc Paulevé.

**Writing – original draft:** Gustavo Magaña-López, Loïc Paulevé.

**Writing – review & editing:** Gustavo Magaña-López, Laurence Calzone, Andrei Zinovyev, Loïc Paulevé.

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
