## [Decision Letter · Decision Letter 0]

12 Mar 2024

Dear Paulevé,

Thank you very much for submitting your manuscript "scBoolSeq: Linking scRNA-Seq Statistics and Boolean Dynamics" for consideration at PLOS Computational Biology.

As with all papers reviewed by the journal, your manuscript was reviewed by members of the editorial board and by several independent reviewers. In light of the reviews (below this email), we would like to invite the resubmission of a significantly-revised version that takes into account the reviewers' comments.

We cannot make any decision about publication until we have seen the revised manuscript and your response to the reviewers' comments. Your revised manuscript is also likely to be sent to reviewers for further evaluation.

Sincerely,

Mark Alber, Ph.D.

Section Editor

PLOS Computational Biology

Mark Alber

Section Editor

PLOS Computational Biology

Reviewer's Responses to Questions

**Comments to the Authors:**

Reviewer #1: Authors propose a novel method for binarizing scRNAseq data to generate a Boolean representation of experimental data. These approaches are crucial for enhancing dynamic modeling of genetic regulatory networks by formalizing data representation. They compare their method to another, presenting advantages. The techniques appear sound, and I believe the work is valuable and merits publication after addressing some queries and making necessary revisions.

Questions:

. Is the classification of scRNAseq data into only three categories sufficiently general?

. The "zeroinf" case appears to be the most noisy. Are the "?" cases discarded? It would be beneficial to observe cases that do not fit into these three categories and how the authors handle them.

. In the second paragraph of the section 'From pseudocounts to...': Can the authors devise a method in scBoolseq to reject a dataset if it is inadequate for binarization? Otherwise, users may employ it in inappropriate cases.

. I was able to install the software without problems, however I did not test it with data.

. A tutorial on using scBoolSeq would be highly beneficial.

. Can an inferred Boolean network be exported from the software to be used in another Boolean tool?

Minor:

Line 38: "active"

Line 57: Please define 'GRN'.

Line 86: "HVGs"

Line 355: Provide a reference for 'k-nearest-neighbors classifier'.

Line 356: Provide a reference for 'by-category bijective matching'.

Please cite the link 'github.com/colomoto/colomoto-docker' for scBoolSeq, as other Colomoto webpages in the web appear to be outdated.

Reviewer #2: In this manuscript, López et al. present scBoolSeq, a method to link scRNA-seq data with Boolean network models by learning gene expression distributions to binarize data or simulate realistic data from Boolean states.

The method addresses an important need and the manuscript is generally well-written. However, the below points should be addressed to strengthen the work. In particular, further evaluation of the binarization results and discussion of computing requirements would help users assess the method's applicability. With these issues addressed, the work will be a valuable contribution to the field.

Major points:

1) How does the binarization model treat housekeeping genes that are constitutively active? By looking at the unimodal example in Figure 3, it seems that these genes will have 3 states instead of being in the 1 state.

2) How do you treat unimodal genes that are not symmetric? Symmetric thresholds may be suboptimal.

3) Regarding the statement on lines 120-121, "The log transformation is necessary in order to ensure the validity of the underlying parametric distributions", please consider briefly expanding on this.

4) Lines 116-117 state "...would represent the standard library size normalisation, yielding counts/reads per million (CPM/RPM)." This is not correct; a multiplicative factor of 1E6 is needed.

5) The text states, "In a first step, genes which do not exhibit a high enough variability". Please define "high enough" quantitatively.

6) Regarding the statement on lines 411-412, "Future engineering work will focus on leveraging the AnnData [59] Python package for handling large datasets that cannot be fit in RAM," please expand on the computing requirements and maximum dataset size that can be analyzed.

Minor points:

It is better to spell "scRNA-Seq" as "scRNA-seq" throughout the manuscript.

On line 127, fix the typo: "patters" -> "patterns".

**Have the authors made all data and (if applicable) computational code underlying the findings in their manuscript fully available?**

Reviewer #1: Yes

Reviewer #2: Yes

PLOS authors have the option to publish the peer review history of their article (what does this mean?). If published, this will include your full peer review and any attached files.

Reviewer #1: No

Reviewer #2: No
---

## [Decision Letter · Decision Letter 1]

24 Jun 2024

Dear Paulevé,

We are pleased to inform you that your manuscript 'scBoolSeq: Linking scRNA-Seq Statistics and Boolean Dynamics' has been provisionally accepted for publication in PLOS Computational Biology.

Best regards,

Christoph Kaleta

Section Editor

PLOS Computational Biology

Mark Alber

Section Editor

PLOS Computational Biology

Reviewer's Responses to Questions

**Comments to the Authors:**

Reviewer #1: The manuscript is now quite clear.

Reviewer #2: The authors did a great job in addressing all the concerns.

**Have the authors made all data and (if applicable) computational code underlying the findings in their manuscript fully available?**

Reviewer #1: Yes

Reviewer #2: Yes

PLOS authors have the option to publish the peer review history of their article (what does this mean?). If published, this will include your full peer review and any attached files.

Reviewer #1: **Yes: **José Carlos Merino Mombach

Reviewer #2: No

---

## [Editor Report · Acceptance letter]

3 Jul 2024

PCOMPBIOL-D-23-01706R1 

scBoolSeq: Linking scRNA-Seq Statistics and Boolean Dynamics

Dear Dr Paulevé,

I am pleased to inform you that your manuscript has been formally accepted for publication in PLOS Computational Biology. Your manuscript is now with our production department and you will be notified of the publication date in due course.

With kind regards,

Olena Szabo
